# Activation of ARP2/3 and HSP70 Expression by Lipoteichoic Acid: Potential Bidirectional Regulation of Apoptosis in a Mastitis Inflammation Model

**DOI:** 10.3390/biom14080901

**Published:** 2024-07-25

**Authors:** Bo Fang, Tingji Yang, Yan Chen, Zhiwei Duan, Junjie Hu, Qi Wang, Yuxuan He, Yong Zhang, Weitao Dong, Quanwei Zhang, Xingxu Zhao

**Affiliations:** 1College of Veterinary Medicine, Gansu Agricultural University, Lanzhou 730070, China; fangb_ioi@163.com (B.F.); ytj19980227@163.com (T.Y.); xyzchenyan@163.com (Y.C.); 15002514846@163.com (Z.D.); hujj@gsau.edu.cn (J.H.); wangqi@gsau.edu.cn (Q.W.); heyuxuan1226@126.com (Y.H.); zhang1234y56@163.com (Y.Z.); 2Gansu Key Laboratory of Animal Generational Physiology and Reproductive Regulation, Lanzhou 730070, China; zhangqw@gsau.edu.cn; 3College of Life Sciences and Biotechnology, Gansu Agricultural University, Lanzhou 730030, China

**Keywords:** mastitis, lipoteichoic acid, ARPC3/4, HSP70, apoptosis

## Abstract

Mastitis typically arises from bacterial invasion, where host cell apoptosis significantly contributes to the inflammatory response. Gram-positive bacteria predominantly utilize the virulence factor lipoteichoic acid (LTA), which frequently leads to chronic breast infections, thereby impacting dairy production and animal husbandry adversely. This study employed LTA to develop models of mastitis in cow mammary gland cells and mice. Transcriptomic analysis identified 120 mRNAs associated with endocytosis and apoptosis pathways that were enriched in the LTA-induced inflammation of the Mammary Alveolar Cells-large T antigen (MAC-T), with numerous differential proteins also concentrated in the endocytosis pathway. Notably, actin-related protein 2/3 complex subunit 3 (ARPC3), actin-related protein 2/3 complex subunit 4 (ARPC4), and the heat shock protein 70 (HSP70) are closely related. STRING analysis revealed interactions among ARPC3, ARPC4, and HSP70 with components of the apoptosis pathway. Histological and molecular biological assessments confirmed that ARPC3, ARPC4, and HSP70 were mainly localized to the cell membrane of mammary epithelial cells. ARPC3 and ARPC4 are implicated in the mechanisms of bacterial invasion and the initiation of inflammation. Compared to the control group, the expression levels of these proteins were markedly increased, alongside the significant upregulation of apoptosis-related factors. While HSP70 appears to inhibit apoptosis and alleviate inflammation, its upregulation presents novel research opportunities. In conclusion, we deduced the development mechanism of ARPC3, ARPC4, and HSP70 in breast inflammation, laying the foundation for further exploring the interaction mechanism between the actin-related protein 2/3 (ARP2/3) complex and HSP70.

## 1. Introduction

Mastitis is classified into two types: contagious and environmental, depending on the bacterial source [1]. Contagious mastitis is frequently caused by Gram-positive bacteria, characterized by robust inflammatory responses and challenging treatment, with *Staphylococcus aureus* (*S. aureus*) being the most prominent pathogen [2]. Infections by *S. aureus* can induce early inflammation; however, they do not effectively activate immune defenses [3]. During this inflammatory response, oxidative stress may lead to cell death, causing membrane rupture and content leakage, which further intensifies the inflammation. Research on the invasion mechanism of Gram-positive bacteria indicated that *S. aureus* indirectly triggered the actin-related protein 2/3 (ARP2/3) complex by binding to fibronectin and host integrin-α5β1, leading to the accumulation of the cytoskeleton at the site of bacterial entry [4]. In addition, the invasion protein INlB from *Listeria monocytogenes* bonded to the cell growth factor receptor MET, activating the verprolin-homologous protein-2 (WAVE2) through the ARP2/3 complex and causing cytoskeleton rearrangement [5,6].

The ARP2/3 complex comprises seven stable peptides: two actin-related proteins (ACTR2 and ACTR3) and five subunits actin-related protein 2/3 complex subunit 1 (ARPC1), actin-related protein 2/3 complex subunit 2 (ARPC2), actin-related protein 2/3 complex subunit 3 (ARPC3), actin-related protein 2/3 complex subunit 4 (ARPC4), and actin-related protein 2/3 complex subunit 5 (ARPC5). ARPC4 is essential for regulating the assembly and stability of the ARP2/3 complex and interacts specifically with ARPC3 [7]. Research indicated that ARPC3 expression was closely linked to cell proliferation, and abnormal levels can disrupt the cell cycle, leading to uncontrolled proliferation [8]. ARPC4 acts as an actin nucleator in the branches of the actin cytoskeleton. Elevated expression enhances cell migration efficiency, thereby accelerating disease progression [9]. The literature has demonstrated that ARPC3 and ARPC4 increased during bacterial invasion, and inhibiting their expression can reduce the severity of the host’s inflammatory response [10,11]. Pathogens that invade bovine mammary epithelial cells triggered the expression of Caspase-8, initiating cell apoptosis pathways [12]. Additionally, the upregulation of NLRP3 can induce autocrine IL-1β, fostering cell metastasis and exacerbating inflammation [13]. Furthermore, ARPC3 and ARPC4 are intricately involved in cell apoptosis. The ARP2/3 complex can initiate cell apoptosis via Clathrin-mediated endocytosis, underscoring its pivotal role [14]. Coincidentally, miR-34a may induce neutrophil apoptosis by regulating F-actin remodeling and generating reactive oxygen species (ROS) through the Cdc42-WASP-ARP2/3 pathway [15]. It has been found that the ARP2/3 complex disrupts mitochondrial distribution and function through Clathrin-mediated endocytosis, resulting in ROS accumulation and oxidative stress, ultimately increasing the early apoptosis rate of goat oocytes [16].

In response to bacterial invasion, the host modulated the expression of various factors to mitigate the inflammatory response, with heat shock protein 70 (HSP70) being particularly significant [17]. HSP70, a highly conserved molecular chaperone protein, was induced under elevated temperatures or stressful conditions. It inhibited the activation of cell apoptosis and inflammatory pathways [18]. Recent study indicated that increased expression of HSP70 can curb ROS-induced cell apoptosis triggered by heat [19]. Furthermore, extensive studies confirmed that HSP70 participated in Clathrin-mediated endocytosis, assisting in vesicle formation and supporting critical cellular functions such as signal transduction [20]. In essence, HSP70 can also facilitate in endocytosis through the neural Wiskott–Aldrich syndrome protein (N-WASP). Additionally, heat shock protein 40 (HSP40), HSP70, and heat shock protein 90 (HSP90) collaborated to regulate the ARP2/3 complex and influence actin polymerization. HSP90 cooperated with phosphorylated N-WASP to modulate actin polymerization mediated by the N-WASP/ARP2-3 complex [21]. During the HSP90-assisted protein maturation process, the newly formed polypeptide first interacts with the HSP40/HSP70 chaperone, then associates with the HSP70 and HSP90 organizing protein (HOP) before being transferred to HSP90 [22]. Moreover, HSP70 can inhibit apoptosis [23]. According to the literature, HSP70 functioned by eliminating intracellular ROS, preventing the release of mitochondrial cytochrome c, and deactivating Caspase3. This process effectively blocked the apoptosis pathway, thereby enhancing shrimp tolerance to air exposure [24].

In prior research, we used lipoteichoic acid (LTA) to induce bovine mammary epithelial cells. The proteomic analysis revealed the upregulation of ARPC3 and ARPC4, alongside alterations in the expression of DNAJB1, a co-chaperone of HSP70, known as HSP40 [25]. Building on these findings, this study aims to examine the effects of the endocytic pathway on inflammation and apoptosis in dairy cow mastitis caused by Gram-positive bacteria, as well as the contributions of these pathways to inflammatory development. Therefore, we used transcriptomic technology to detect and analyze the mammary alveolar cells-large T antigen (MAC-T) inflammation model. The virulence factor LTA from *S. aureus* was used to induce inflammation in both bovine mammary epithelial cells and a mouse model, with alterations in protein levels associated with the endocytic, inflammatory, and apoptotic pathways validated through molecular biology techniques. This research will establish a solid foundation for future studies, elucidating the regulatory roles of key factors ARPC3, ARPC4, and HSP70 in dairy cow mastitis, thereby facilitating the identification of new therapeutic targets for this condition.

## 2. Materials and Methods

### 2.1. Cell Culture and Treatment

The MAC-T cells, sourced from the Chinese Academy of Agricultural Sciences, were cultured in Gibco DMEM/F12 medium (Gibco, New York, NY, USA) supplemented with 10% fetal bovine serum (Invigentech, Irvine, CA, USA) and maintained at 37 °C in a 5% CO_2_ incubator. Upon reaching the logarithmic growth phase, MAC-T cells were planted into six-well plates using 0.25% trypsin-EDTA (Gibco, New York, NY, USA). Samples were collected once the cells attained approximately 70% confluence after treatment with 10 μg/mL LTA (Sigma, St. Louis, MO, USA) for 24 h, designated as the 10 group, with the untreated cells serving as the control (Con group), in accordance with the protocol outlined in W. Dong’s article [25].

### 2.2. Transcriptome Analysis

MAC-T cells were treated with 0 and 10 μg/mL LTA at a density of 1 × 10^7^, and samples were collected after 24 h of treatment with 3 replicates per group. Total RNA was extracted using TRNzol Universal reagent (TIANGEN, Beijing, China), and its quantity and purity were assessed using an Ultramicro biochemical spectrophotometer (NanoDrop ND-1000, Thermo Fisher, San Diego, CA, USA). High-quality RNA samples were selected for library construction. Subsequently, 1 µg of total RNA was purified and fragmented, followed by reverse transcription into cDNA. The fragmented RNA was then subjected to A-tailing of the blunt ends of each strand with adapters. The ligated products underwent PCR amplification at 20 °C for 15 min, followed by purification. Finally, the results were quantified, inspected, and tested using computer analysis (4200 TapeStation, Agilent Technologies, Santa Clara, CA, USA). Gene set enrichment analysis (GSEA) was performed to rank all genes based on theirdifferential expression levels (log_2_FC) between the two sample groups, after the Kyoto Encyclopedia of Genes and Genomes (KEGG) pathway enrichment analysis. Differential factors were used for predicting protein interactions using the STRING website (STRING: functional protein association networks (https://cn.string-db.org/cgi/network?taskId=bTL9BgMxrvuR&sessionId=bsDo2zYkNAp1 (accessed on 25 April 2024)).

### 2.3. Mammary Gland Model Construction

Fifty 8-week-old female Kunming mice and 25 male Kunming mice were obtained from the Chinese Academy of Agricultural Sciences. After a week of acclimatization, they mated in a 2:1 ratio of male to female for two days. The 18 pregnant mice with similar physical conditions were selected and divided into two groups after 7 days of breastfeeding. Then, the milk ducts of the fourth pair of the mice’s mammary glands were injected. One group was injected with 50 μL of normal saline as a control, named the Con group, and the other group was injected with 50 μL of 20 μg/mL of LTA, named the CM group. Twenty-four hours after the mice were injected, a fourth pair of mammary tissue was collected under sterile conditions. Some were fixed in 4% formaldehyde solution and the other were stored at −80 °C for future use. All samples were collected following the ethical guidelines approved by the Animal Care Commission of Gansu Agricultural University.

### 2.4. Immunological Staining

Mouse mammary tissue preserved in formaldehyde solution was sectioned into 4 μm thick sections and subjected to hematoxylin–eosin staining (HE) after deparaffinization in xylene and graded alcohols. Following staining, sections were dehydrated, sealed, and observed under a Zeiss microscope (Axiocam 208 color, Zeiss, Oberkochen, Germany). Immunohistochemistry (IHC) was performed using the ABC system, followed by heat antigen retrieval and blocking. Primary antibody incubation was performed overnight at 4 °C, after one hour of secondary antibody incubation, DAB staining, hematoxylin counterstaining, and mounting. Images were captured using the same microscope camera system. For multi-color fluorescent staining, sections were incubated with secondary antibodies using a three-marker four colors multiple fluorescent staining kit (AiFang Biological, Changsha, China) at 37 °C for 1 h in the dark. After washing, sections undergo repeated incubation with subsequent primary antibody-HRP secondary antibody pairs, followed by additional tyramide fluorescent substrate. Finally, sections were mounted with an anti-fluorescence quenching mounting medium, and images were captured using an inverted fluorescence microscope (Revolve Omega, apexbio, Suzhou, China). Cell samples were fixed in 4% paraformaldehyde, permeabilized with 0.3% Triton X-100, blocked for 1 h, and underwent similar subsequent steps as tissue sections.

### 2.5. TUNEL Staining and Flow Cytometry

Cells were initially washed with PBS and then fixed with paraformaldehyde. Following fixation, 0.3% Triton X-100 was added, and the cells were allowed to incubate at room temperature for 5 min. A mixture of TdT enzyme (Roche, 11684795910, Basel, Switzerland) and fluorescent labeling solution, at a ratio of 1:9, was prepared and 50 μL of this mixture was added to a six-well plate. The plate was then incubated at 37 °C in the dark for 60 min. Subsequently, the cells were mounted using the anti-fluorescence quenching mounting medium, and images were captured using an inverted fluorescence microscope (Revolve Omega, apexbio, Suzhou, China). Meanwhile, the cells were digested, centrifuged, and gently resuspended in PBS. They added a mixture of 195 μL Annexin V-FITC conjugate and 5 μL Annexin V-FITC (BD, San Jose, CA, USA) and gently mixed it with 10 μL of propidium iodide staining solution. The samples were then incubated in the dark at room temperature for 10–20 min. Finally, they were stored on ice and analyzed using a flow cytometer (CytoFlex, Beckman Coulter, CA, USA).

### 2.6. RNA Isolation, cDNA Synthesis, and qPCR

Total RNA was extracted from the mouse mammary glands and cells using the Trizol reagent kit (Solarbio, Beijing, China), followed by cDNA synthesis using the Evo M-MLV RT Kit (Agbio, Changsha, China). Real-time quantitative PCR detecting system (qPCR) was conducted using 2× SYBR^®^ Green Pro Taq HS Premix (Selleck, Los Angeles, CA, USA) according to the manufacturer’s instructions and performed using the Light Cycler 96 real-time system (Roche, Basel, Switzerland), and the results were analyzed using the 2^−∆∆CT^ method to determine the relative expression of the target genes. GAPDH was amplified in parallel as an internal control.

### 2.7. Western Blot

The relative protein expression levels of IL-1β, IL-6, ARPC3, ARPC4, HSPA1A, HSPA1L, Bax, Bcl-2, Caspase7 (proteintech, Wuhan, China), TNF-α, Caspase3, and Caspase8 (Bioss, Beijing, China) in the mammary gland tissues and MAC-T were measured using Western blot. Total protein was extracted from mammary gland tissue/cells using cold RIPA buffer (Solarbio, Beijing, China). Primary antibodies were incubated overnight at 4 °C, followed by incubation with corresponding secondary antibodies (goat anti-rabbit IgG Secondary antibody, SAB, Greenbelt, MD, USA; goat anti-mouse IgG Secondary antibody, SAB, Greenbelt, MD, USA) at 37 °C for 1 h. Band densitometry was quantified and analyzed using Image-J 1.52a.

### 2.8. Statistical Analysis

Statistical analyses were conducted using SPSS 22.0 software. The test data were presented as “mean ± standard deviation”. One-way analysis of variance (ANOVA) was utilized for comparisons among multiple groups, followed by post hoc least significant difference (LSD) tests for pairwise comparisons. Prior to ANOVA, the statistical data were assessed for normality, confirming that all data met the criteria for normal distribution. A significance level of *p* < 0.05 indicated a significant difference, while *p* < 0.01 indicated an extremely significant difference. The analysis results were visualized by using GraphPad Prism 9.0 software.

## 3. Results

### 3.1. Induction of Inflammation and Activation of Related Inflammatory Factors by LTA

Following treatment with LTA, the analysis of mouse mammary gland tissues using HE and IHC showed that the Con group had enlarged glandular follicles, with neatly organized mammary epithelial cells and no inflammatory cell infiltration (Figure 1A). In contrast, the CM group exhibited ruptured glandular follicles, the detachment of some mammary epithelial cells, and the infiltration of numerous inflammatory cells into the follicles. IHC analysis confirmed a significant increase in the inflammatory cytokines TNF, IL-1β, and IL-6 (*p* < 0.01) (Figure 1B). Additionally, qPCR and Western blot results from mouse mammary gland tissues and MAC-T cells after LTA treatment showed a significant increase in TNF, IL-1β, and IL-6 compared to the control group (*p* < 0.01) (Figure 1C–E). These findings suggested that LTA treatment enhances the expression of inflammatory cytokines in mouse mammary glands and MAC-T cells, leading to tissue damage and indicating the successful establishment of a mastitis model induced by LTA. Please see the original Western blot image in the Appendix A. 

### 3.2. Transcriptomic Analysis Reveals Expression of Related Factors in LTA-Activated Endocytosis and Apoptotic Pathways

Transcriptomic analysis was conducted on MAC-T cells treated with Con and 10 μg/mL LTA. The transcriptomic data were subjected to GSEA to identify enriched pathways using the KEGG database. The analysis revealed the enrichment of 20,346 genes across 340 pathways. Specifically, 96 genes were enriched in the endocytic pathway, while 24 genes were enriched in the apoptotic pathway (Figure 2A,B). In the endocytic pathway, we observed the significant upregulation of *ARPC3*, *ARPC4*, and *HSP70*. Similarly, in the apoptotic pathway, *HSP70*, *Caspase9*, and *Bax* exhibited upregulated expression. These findings suggested that *HSP70* was simultaneously involved in both pathways. Using the STRING website (https://string-db.org/, accessed on 25 April 2024), we generated a predicted protein–protein interaction network (Figure 2C). The network diagram illustrated that *ARPC3* interacted with *ARPC4*, mediating the action of *HSP70* (*HSPA1L*, *HSPA1A*) on apoptosis-related factors *Caspase3* and *Caspase7*.

### 3.3. Validating Gene Expression in Inflammation Models and Assessing Cell Apoptosis Based on Transcriptomic Analysis

In addition, qPCR and Western blot were carried out to assess the expression of ARPC3, ARPC4, HSPA1A, and HSPA1L proteins in mouse mammary gland tissues (Figure 3A,B). Results illustrated a marked elevation in mRNA and protein expression in the CM compared to the Con group. We employed immunofluorescence staining to visualize the localization of ARPC3, ARPC4, and HSP70 proteins in mammary gland tissues and cells. These proteins were observed in the cytoplasm. The immunofluorescence signal intensity for ARPC3, ARPC4, and HSP70 was significantly greater in the CM group than in the Con group (*p* < 0.01) (Figure 3C). mRNA expression levels of *ARPC3*, *ARPC4*, *HSPA1A*, and *HSPA1L* in MAC-T were evaluated and revealed a notable increase in mRNA expression in the 10 group compared to the Con group (Figure 3D). Western blot (Figure 3E) and immunofluorescence (Figure 3F) showed the same consequence in mouse mammary tissue.

Apoptosis was determined by TUNEL and flow cytometry. Both techniques revealed a significant increase in the apoptosis rate in the 10 group compared to the Con group, as well as in the CM group compared to the Con group (Figure 4A,B). qPCR and Western blot were evaluated for Caspase3, Caspase7, Caspase8, Bcl-2, and Bax in mouse mammary tissue (Figure 5A,B) and MAC-T (Figure 5C,D). In comparison to the Con group, Bcl-2 expression was significantly reduced, while the others showed a significant increase. Consistent results were noted in both the MAC-T inflammation model and mouse mammary gland tissues.

## 4. Discussion

*S. aureus* is recognized as one of the primary pathogens contributing to mastitis in cows [2,26]. LTA serves as a key constituent of the cell wall in Gram-positive bacteria, capable of triggering immune network pathways in host cells. This activation leads to oxidative stress, autophagy, and apoptosis in mammary epithelial cells. These attributes position LTA as a preferred agent for modeling Gram-positive bacterial inflammation [27,28,29]. Consequently, we established mastitis models in both MAC-T and mice using LTA, aiming to delve deeper into strategies for the prevention and treatment of bovine mastitis.

The onset of mastitis initiates an immune response in mammary tissue, triggering a cascade of reactions. *S. aureus* exploits this response by disrupting normal cellular functions in the tissue, facilitating its invasion [30]. In our previous research, proteomic analysis unveiled that LTA induces mastitis by modulating ARPC3 and ARPC4, thereby activating the host immune response [25]. In this study, we employed GSEA and KEGG pathway enrichment, identifying 167 upregulated and 174 downregulated pathways. These pathways predominantly relate to cellular phagocytosis, inflammatory responses, and cellular processes like the cell cycle and the NF-κB (nuclear factor kappa B) pathway. We focused on IL-1β and TNF-α, which were linked to endocytosis and apoptosis. Integrating previous research showing that TNF-α can induce apoptosis [31], we confirmed their varying expression patterns in the inflammation model.

IHC and IF results confirmed that ARPC3 and ARPC4 proteins were mainly located on the cell membrane (Figure 3C,F), which also proved that they led the endocytic pathway. Andrew, M., et al. discovered that *S. aureus* induced actin rearrangement in host cells to achieve complete internalization in a time-, dose-, and temperature-dependent manner [32]. Additionally, *S. aureus* invasion activated the ARP2/3 complex via Clathrin-mediated endocytosis, aiding the internalization, dissemination, and evasion of host immune responses [33]. Research indicated that the activation of the ARP2/3 complex upon bacterial invasion accelerates bacterial colonization within cells [34]. We also found a significant increase in the expression of ARPC3 and ARPC4 through protein expression analysis. Similarly, in *Shigella flexneri*, actin nucleation at the bacterial pole relied on the recruitment of the nucleation-promoting factor N-WASP, which activated the actin-nucleating agent ARP2/3. Cell-to-cell dissemination occurred through actin-based movement and the formation of membrane prominences at cell-to-cell contact in the colon epithelium. These protrusions ruptured into neighboring cells, breaking down into vacuoles from which pathogens escape [35]. In addition, studies manifested that the ARP2/3 complex cooperated with the nucleation-promoting factors of the WASP family to regulate inflammation and Caspase9 expression, participating in the apoptosis mechanism [36,37,38]. In WASP family genes, JMY plays a crucial role in mediating apoptosis, which necessitates actin nucleation by the ARP2/3 complex. Actin filament assembly occurs in cytochrome C clusters, with caspase 3 present in the cytoplasmic region [39]. Evidence suggested that cofilin and ARP2/3 were localized to vesicles associated with apoptosis. Hence, ARP2/3 complexes were key participants in apoptosis [16,40]. On this basis, we proposed that LTA activates the endocytic pathway, which increased the expression of ARPC3 and ARPC4, leading to cell apoptosis.

HSP70 served as a pivotal regulatory mechanism in host immune protection, exhibiting heightened expression during inflammation to trigger the host’s immune response and impede cell apoptosis. Integrating these findings, LTA invasion activated ARP2/3 through the endocytic pathway, causing cytoskeletal reorganization. This facilitated the entry of numerous virulence factors, culminating in cellular inflammation and apoptosis. The interplay between these factors disrupted the cellular environment. To evade self-destruction and bolster host immune defenses, the host elevated HSP70 expression, facilitating the lysosomal degradation of virulence factors and inhibiting apoptosis. Previous studies indicated that the upregulation of HSP70 can impact the formation of actin-myosin filaments in experimental stroke [41], suggesting potential interactions between HSP70 and actin as well as cytoskeleton-related proteins. Interestingly, both ARPC3/4 and HSP70 were enriched in the Clathrin-mediated endocytic pathway. From the pathogen’s perspective, invasion triggered certain pathogen proteins to bind to cell receptors, while others were engulfed into cells via phagosomes, inducing mitochondrial dysfunction and subsequently activating cellular inflammation and apoptosis [42]. From the host cell’s perspective, pathogen invasion activated the immune system, prompting cellular phagocytosis of pathogens, followed by lysosomal degradation to prevent adverse reactions like cell apoptosis [43]. 

The results indicated that LTA induced cell apoptosis, prompting the host to initiate an innate immune response by upregulating HSP70 expression, which helped mitigate inflammation-related damage from apoptosis. Further validation indicated a potential bidirectional regulatory mechanism of the endocytic pathway in the cell apoptosis response triggered by inflammatory reactions. So, we made a mode pattern about it (Figure 6). The pattern showed when *S. aureus* invaded cells, virulence factors were internalized via endocytosis, leading to the increased expression of ARPC3/4 and N-WASP. HSP70-targeted ARPC3/4 released the virulence factor and finally caused cell apoptosis. Simultaneously, HSP70 interacted with N-WASP through HSP90, which inhibited JNK, thereby reducing apoptosis. This interaction led to the upregulation of Bax, Caspase3, Caspase7, and Caspase8, while downregulating Bcl-2 and activating the innate immune response. As the crucial step of endocytosis and apoptosis, the role of HSP70, and the relation between HSP70 and ARPC3/4 was still indistinct. However, due to the limitations of the current experiments, we lack direct evidence to prove the straightforward relationship between ARPC3/4 and HSP70, and the regulatory role of the three on apoptosis. Follow-up studies to explore this mechanism more comprehensively are in progress.

## 5. Conclusions

In LTA-induced mammary inflammation, we identified 120 genes enriched in the endocytic and apoptotic pathways of interest. Molecular verification revealed a positive correlation between endocytic pathway-related proteins ARPC3 and ARPC4 and the occurrence of inflammation. Additionally, pro-apoptotic factors showed a positive correlation with inflammation. Interestingly, the expression of HSP70, which was associated with immune regulation, was also positively correlated. This led us to speculate whether increased HSP70 expression could inhibit ARPC3 and ARPC4, thus protecting cellular homeostasis. Consequently, this study inferred that the endocytic pathway played a vital role in udder infection in dairy cows. These findings are anticipated to enhance our understanding of the relationship between inflammation and apoptosis in bovine mastitis, offering valuable insights into its prevention and treatment.

## Figures and Tables

**Figure 1 biomolecules-14-00901-f001:**
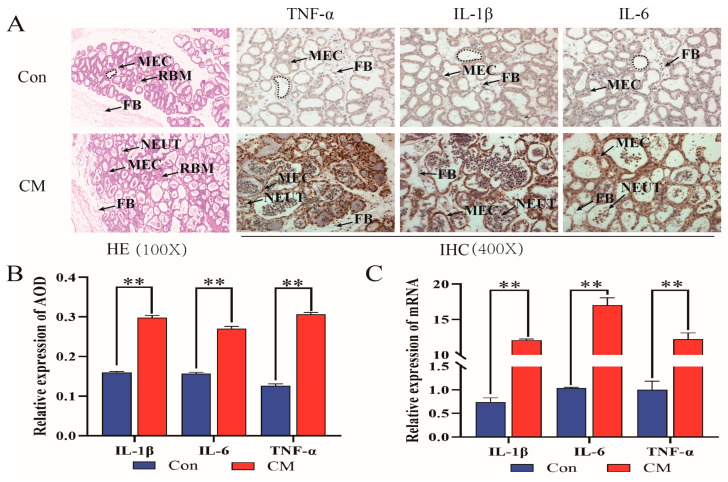
High expression of TNF-α/IL-1β/IL-6 accelerates the inflammation process in vivo and in vitro. (**A**) The neutrophil infiltration in HE staining and TNF-α/IL-1β/IL-6 IHC staining. FB, fibroblast; MEC, mammary epithelial cell; NEUT, neutrophil. The IHC panel is a high magnification (400×, scale 50 microns) image that corresponds to the HE panel (100×, scale 200 microns). (**B**) IHC scores of each group. (**C**,**D**) Quantification of relative mRNA expression of *TNF-α*, *IL-1β*, and *IL-6*, in mouse mammary tissue (**C**) and MAC-T (**D**). (**E**) Relative expression of TNF-α, IL-1β, and IL-6 protein in mouse mammary tissue. Data are means ± SEM (*n* = 3 per group). **, *p* < 0.01. Please see the original Western blot image in the Appendix A.

**Figure 2 biomolecules-14-00901-f002:**
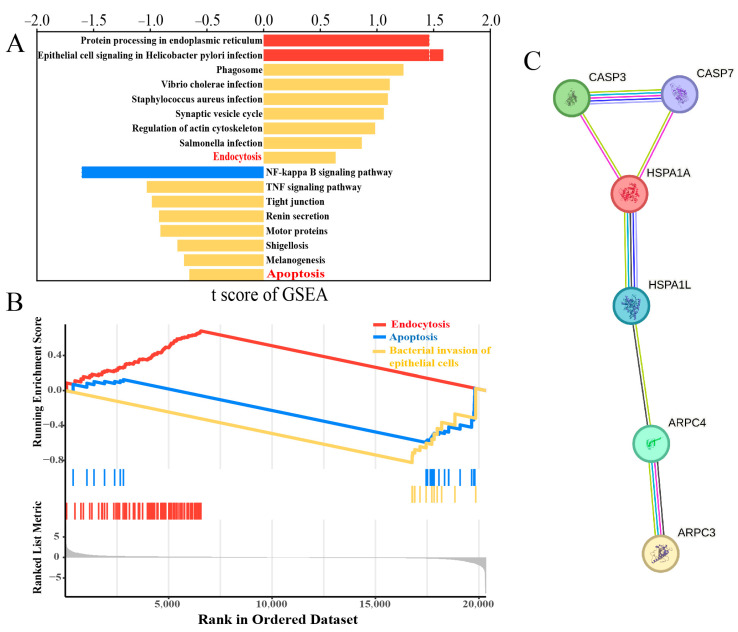
Analysis of transcriptomic data processed by LTA. (**A**) Partial pathway of GSEA. (**B**) Endocytosis, apoptosis, and *S. aureus* infection were enriched by GSEA. (**C**) The relationship predicted by STRING.

**Figure 3 biomolecules-14-00901-f003:**
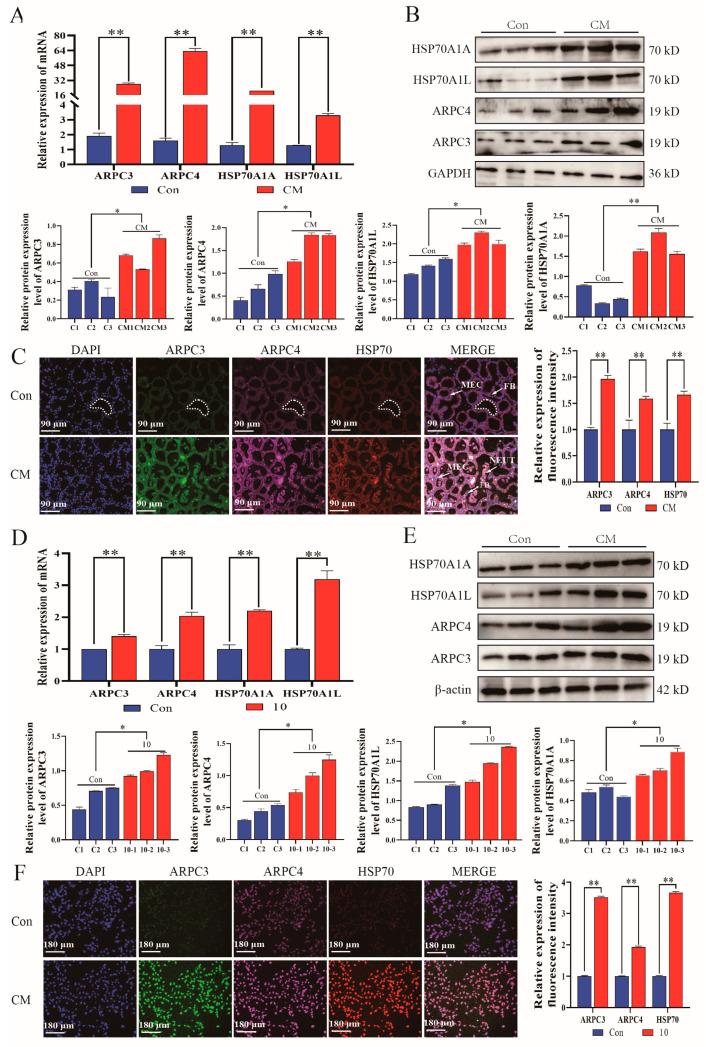
Inflammation enhanced the expression of ARPC3/ARPC4/HSP70. (**A**,**B**) Relative expressions of mRNA (**A**) and protein (**B**) of ARPC3, ARPC4, and HSP70 in mouse mammary tissue. (**C**) Colocation of ARPC3, ARPC4, and HSP70 in mouse mammary tissue. (**D**,**E**) Relative expression of mRNA (**D**) and protein (**E**) of ARPC3, ARPC4, and HSP70 in MAC-T. (**F**) Colocation of ARPC3, ARPC4, and HSP70 in MAC-T. Data are means ± SEM (*n* = 3 per group). *, *p* < 0.05; **, *p* < 0.01. Please see the original Western blot image in the Appendix A.

**Figure 4 biomolecules-14-00901-f004:**
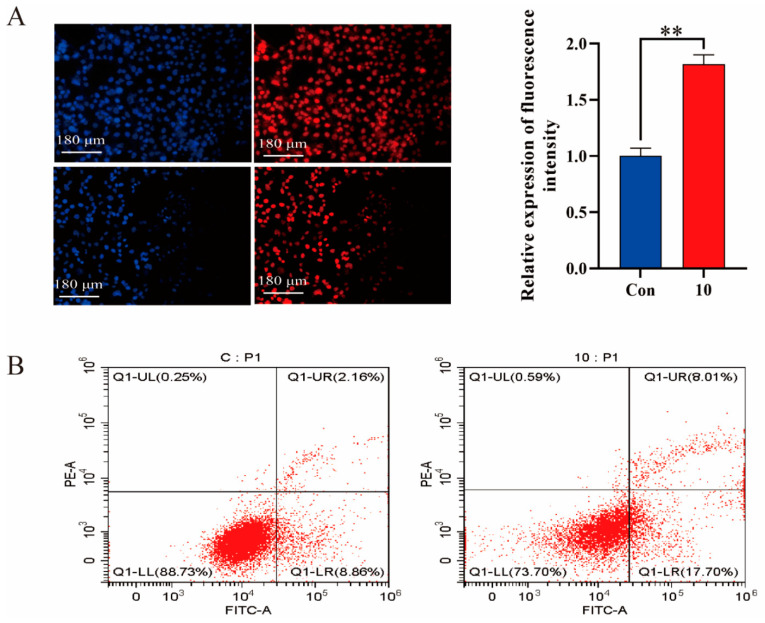
Apoptosis detection by TUNEL and FC. (**A**) Apoptosis detection by TUNEL in MAC-T. (**B**) Apoptosis assay by FC in MAC-T. **, *p* < 0.01.

**Figure 5 biomolecules-14-00901-f005:**
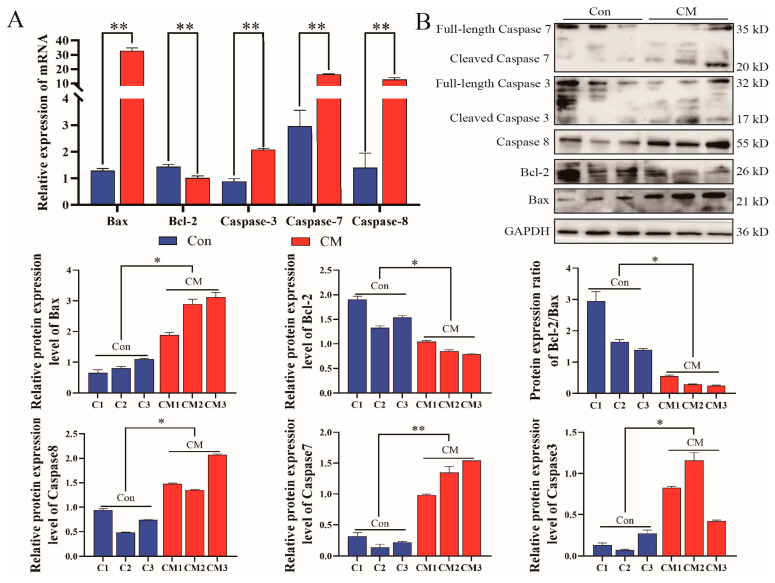
Apoptosis was increased under the effect of LTA. (**A**) Quantification of relative mRNA expression of *Caspase3*, *Caspase7*, *Caspase8*, *Bax*, and *Bcl-2* in mouse mammary tissue. (**B**) The relative expression of Caspase3, Caspase7, Caspase8, Bax, and Bcl-2 protein in mouse mammary tissue. (**C**) Quantification of relative mRNA expression of *Caspase3*, *Caspase7*, *Caspase8*, *Bax*, and *Bcl-2* in MAC-T. (**D**) Relative expression of Caspase3, Caspase7, Caspase8, Bax, and Bcl-2 protein in MAC-T. Data are means ± SEM (*n* = 3 per group). *, *p* < 0.05; **, *p* < 0.01. Please see the original Western blot image in the Appendix A.

**Figure 6 biomolecules-14-00901-f006:**
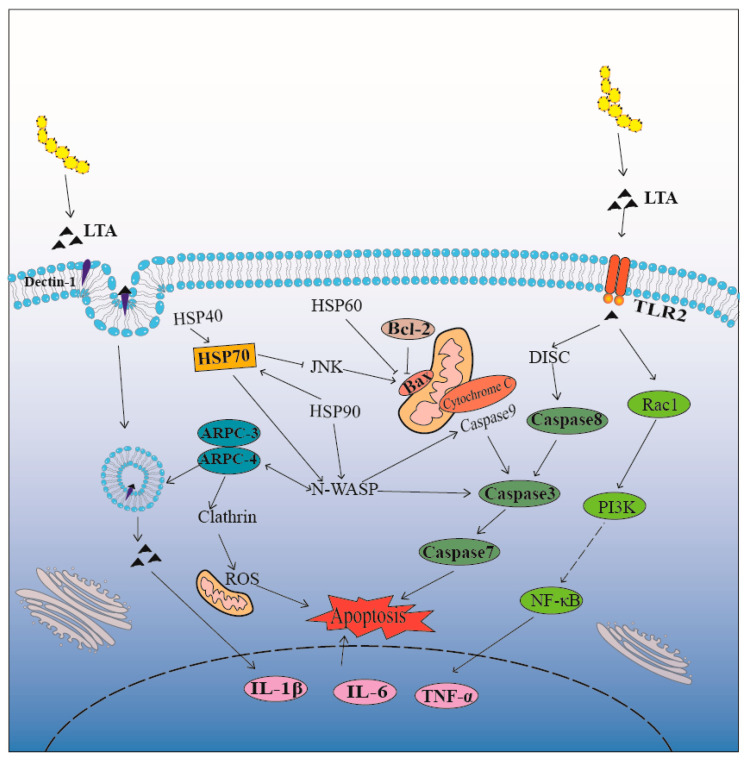
Model pattern of LTA-induced apoptosis via ARPC3/ARPC4/HSP70. LTA modulated ARPC3 and ARPC4 expression in the endocytic pathway inducing cell apoptosis and activating HSP70 to mitigate sustained host cell innate immunity. In addition, LTA can also induce the inflammatory response by activating the TLR2/PI3K-AKT/NF-κB pathway.

## Data Availability

The datasets analyzed or generated during the study are available from the authors (d.wt2008@163.com).

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
