# Peer review of "Activation of ARP2/3 and HSP70 Expression by Lipoteichoic Acid: Potential Bidirectional Regulation of Apoptosis in a Mastitis Inflammation Model"

_biomolecules, 2024, doi:10.3390/biom14080901_

Round 1

Reviewer 1 Report

Comments and Suggestions for Authors

・The authors described that S. aureus LTA upregulateed proinflammatory cytokines, actin-related protein 2/3 complex subunits, HSP70, and apoptosis-promoting factors. However, the authors did not show interrelationship between each event. This manuscript is a just enumeration of biological responses induced by S. aureus LTA.

・Letters in Figures are too small to recognize even after enlargement. Please use fonts, which are well legible.

・Most of references lack page numbers and article numbers.

Specific comments

・“Staphylococcus aureus” should be in italic. “Staphylococcus aureus” should be ”S. aureus (in italic)” after second appearance.

・Line 17-18: “HSP40 ligand of the HSP70” is wrong. HSP40 is a cochaperone of HSP70.

・Line 78-79: “DNAJB1 of HSP70” is wrong. DNALB1 is a member of HSP40, which is a cochaperone of HSP70.

・Figure 1B: What is “AOD”?

・Figure 1E and 5B and D: What do you mean ”optical density”? These seem to be band intensity of Western blotting.

・Line 248-255: The sentence “Consistent results were observed ------- mouse mammary gland tissues” should be moved last of this paragraph.

・Section 3.4 and Figure 6 are a just speculation. The description should be moved to Discussion section, but not Results section.

・Section 3.4 and Figure 6: Toll-like receptor 2 pathway is most popular and important pathway of causing proinflammatory response by LTA. This lacks in authors’ speculation.

Comments on the Quality of English Language

Neumerous complicated sentences and gramatical errors are found in this manuscript. 

Author Response

Comments 1: The authors described that S. aureus LTA upregulated proinflammatory cytokines, actin-related protein 2/3 complex subunits, HSP70, and apoptosis-promoting factors. However, the authors did not show interrelationship between each event. This manuscript is a just enumeration of biological responses induced by S. aureus LTA.

Response 1: Thank you for pointing this out. In this article, we extracted and verified the pathways and genes we were interested in through transcriptomics data. Due to experimental limitations, the specific relationship between ARPC3/ARPC4 and HSP70 cannot be fully explained, but their relationship was deduced in the discussion section of the article, lines 351 to 366, page 10. And we have already carried out follow-up work to verify this interaction.

Comments 2: Letters in Figures are too small to recognize even after enlargement. Please use fonts, which are well legible.

Response 2: Agree. Thanks for your careful checks. We have checked all figures in the manuscript, and have uploaded clearer figure on page 8 again.

Comments 3: Most of references lack page numbers and article numbers.

Response 3: Agree. As suggested by the reviewer, we have completed the references pages and corrected the reference format in lines 397 to 497, pages 13 to 15.

Comments 4: “Staphylococcus aureus” should be in italic. “Staphylococcus aureus” should be” S. aureus (in italic)” after second appearance.

Response 4: Agree. We feel sorry for our carelessness. We have revised the incorrect writing format in lines 39 and 40, page 1; line 45, page 2; line 101, page 3; line 246, page 7; lines 287 and 296, page 10; lines 309, 311 and 356, page 11; line 355, page 11.

Comments 5: Line 17-18: “HSP40 ligand of the HSP70” is wrong. HSP40 is a cochaperone of HSP70.

Response 5: Agree. Thanks for your insightful vision. Because of rewriting the abstract, and missing the sentence, we have changed the same mistake in lines 95 to 96, page 2.

Comments 6: Line 78-79: “DNAJB1 of HSP70” is wrong. DNALB1 is a member of HSP40, which is a cochaperone of HSP70.

Response 6: Agree. Thank you for your professional reminder. We have modified the sentence in lines 95 to 96, page 2.

Comments 7: Figure 1B: What is “AOD”?

Response 7: Thank you again for your question which made us realize our mistake. In our opinion, AOD is short for average optical density, and the date is obtained from Image J scanning IHC film optical density. But on account of cursoriness, we didn’t write the full name.

Comments 8: Figure 1E and 5B and D: What do you mean “optical density”? These seem to be band intensity of Western blotting.

Response 8: When we perform Western blotting band analysis, we will scan the optical density of the band, and the scanning result is the optical density. We determine the relative expression of proteins by analyzing the optical density.

Comments 9: Line 248-255: The sentence “Consistent results were observed ------- mouse mammary gland tissues” should be moved last of this paragraph.

Response 9: Agree. As you suggested, we have deleted the sentence.

Comments 10: Section 3.4 and Figure 6 are a just speculation. The description should be moved to Discussion section, but not Results section.

Response 10: Agree. Following your advice, we have moved Figure 6 to discussion.

Comments 11: Section 3.4 and Figure 6: Toll-like receptor 2 pathway is most popular and important pathway of causing proinflammatory response by LTA. This lacks in authors’ speculation.

Response 11: Thank you for your very professional vision. Other members of the research group have published a paper in the International Journal of Molecular Sciences entitled Toll-like Receptor 2 Is Associated with the Immune Response, Apoptosis, and Angiogenesis in the Mammary Glands of Dairy Cows with Clinical Mastitis. Therefore, this article did not repeat the experimental work. However, I have supplemented Figure 6 based on the previously published paper.

4. Response to Comments on the Quality of English Language

Point 1: Numerous complicated sentences and grammatical errors are found in this manuscript.

Response 1: We tried our best to improve the manuscript and made some changes to the manuscript. These changes will not influence the content and framework of the paper. We also have engaged a native English speaker to thoroughly revise our language, ensuring the fluency, accuracy, and consistency of our writing. We appreciate for Reviewers’ warm work earnestly and we hope the revised manuscript will be acceptable to you.

Reviewer 2 Report

Comments and Suggestions for Authors

Dear authors,

It was my pleasure to review the manuscript entitled "Activation of ARP2/3 and HSP70 Expression by Lipoteichoic Acid: Potential Bidirectional Regulation of Apoptosis in a Mastitis Inflammation Model". This paper has an interesting topic since the bacterial infections are very common and represent challenge for treatment. I am recommending the manuscript for publication after minor changes. My comments are listed below:

- The novelty of the study should be more prominent in the introduction

- Line 90: Rewrite the sentence

- Line 116: Rewrite the sentence

- How many replicates of experiments

- in vitro - Italic

-Figure 2: S. aureus in Italic

-Rewrite the conclusion

Author Response

Comments 1: The novelty of the study should be more prominent in the introduction.

Response 1: Thank you for pointing this out. We agree with this comment. Therefore, I/we have rewritten the introduction in lines 37 to 107, pages 1 to 3.

Comments 2: Line 90: Rewrite the sentence.

Response 2: Agree. We were sorry for our careless mistakes. Thank you for your reminder. We have revised the sentence in line 110, page 3.

Comments 3: Line 116: Rewrite the sentence.

Response 3: Agree. As suggested by the reviewer, we have revised it in line 136, page 3.

Comments 4: How many replicates of experiments?

Response 4: For the in vitro experiment, we had 3 independent replicates per group. For the in vivo experiment, we had 9 independent replicates. These 9 independent replicates were first mixed before qPCR and western blotting experiments, and then evenly divided into 3 new replicate samples.

Comments 5: in vitro – Italic.

Response 5: Agree. We feel sorry for our carelessness. Thanks for your correction. We have revised the mistake in line 223, page 6. Not only that, we also changed in vivo to italics in line 222, page 6.

Comments 6: Figure 2: S. aureus in Italic.

Response 6: Agree. Thanks for your careful checks. We have revised the mistake in line 246, page 7. Not only that, we also changed S. aureus in another place to italics.

Comments 7: Rewrite the conclusion

Response 7: Agree. We think this is an excellent suggestion. We have modified the conclusion to emphasize the results obtained through our work content and the inferences based on the results. The modified content is in lines 372 to 383, page 12.

Reviewer 3 Report

Comments and Suggestions for Authors

Overall the paper presents interesting results and the experiments are well designed.

I recommend its acceptance.

Author Response

Thank you for your comments concerning our manuscript entitled “Activation of ARP2/3 and HSP70 Expression by Lipoteichoic Acid: Potential Bidirectional Regulation of Apoptosis in a Mastitis Inflammation Model” (ID: biomolecules-3061321). We are delighted by your comments, but we have also made some changes based on the comments of other reviewers. We have studied comments carefully and have made corrections which we hope meet with approval. Revised portions are marked in red on the paper. Thank you again for your comments on our article.

Reviewer 4 Report

Comments and Suggestions for Authors

-The Introduction needs to be shortened to become more concise.

-The objectives must be defined clearly.

-Please underline the gaps in the literature that will be filled by this publication.

-Please add a new sub-section: 2.9. Control. Please include in there all controls use din this study: materials, chemicals, mice, bacterial isolates, mammary glands within animals. All the controls must be defined clearly and described separately.

-Analysis. Before carrying out analysis with ANOVA, please demonstrate that data had a normal distribution. Otherwise, please use non-parametric tests.

-Comment about visualization. Visualization is good. No need for change.

-Comment about tables. The authors can increase the number of tables in the revised document and reduce the length of the text.

-Please include a paragraph about commercial extension of this work. Do you think that the new findings will be available commercially before the end of the year?

-References. The authors have missed some important references, so please add these and discuss them in relation to the findings.

-Conclusion. These are a bit vague, so please make it more concise and more relevant to the findings of the present study.

Recommendation. Revision and re-evaluation.

Author Response

Comments 1: The Introduction needs to be shortened to become more concise.

Response 1: Thank you for pointing this out. We agree with this comment. Therefore, I/we have rewritten the introduction in lines 37 to 107, pages 1 to 3.

Comments 2: The objectives must be defined clearly.

Response 2: Agree. Your comments are very pertinent. We have revised the article repeatedly and modified the discussion and conclusion sections on pages 10 to 13 to highlight our objectives.

Comments 3: Please underline the gaps in the literature that will be filled by this publication.

Response 3: Agree. Thank you for your careful reading of the article. We are very grateful and agree with your comments. To emphasize the highlights of the article, we have revised the discussion section in 350 to 365, pages 12 to 13. Further, reflect the gaps filled by this article.

Comments 4: Please add a new sub-section: 2.9. Control. Please include in there all controls used in this study: materials, chemicals, mice, bacterial isolates, mammary glands within animals. All the controls must be defined clearly and described separately.

Response 4: Due to the structure of our article, we have modified the Materials and Methods section to highlight our control group. The cow mammary cell control group is in lines 110 to 118, page 3; the mouse control group is in lines 136 to 146, page 3. For all reagents and drugs in this experiment, we have marked them where they appear.

Comments 5: Analysis. Before carrying out analysis with ANOVA, please demonstrate that data had a normal distribution. Otherwise, please use non-parametric tests.

Response 5: Agree. We conducted a normal analysis of experimental data before conducting an ANOVA analysis but omitted the normal analysis when writing. The data all conform to the normal distribution, so the ANOVA method is used for analysis.

Comments 6: Comment about visualization. Visualization is good. No need for change.

Response 6: Thank you for your laud. In the future scientific research life, we will strive for excellence.

Comments 7: Comment about tables. The authors can increase the number of tables in the revised document and reduce the length of the text.

Response 7: Thank you very much for your careful review and suggestions. However, after our consideration, we all agree that text is less space-consuming in short texts. As far as this manuscript is concerned, we have not found any place where a table can be used instead.

Comments 8: Please include a paragraph about commercial extension of this work. Do you think that the new findings will be available commercially before the end of the year?

Response 8: Thank you for your advice. Since the experiments we are currently doing cannot support our commercialization, we are currently further advancing the depth of the experiments based on this article, to finally complete the commercialization.

Comments 9: References. The authors have missed some important references, so please add these and discuss them in relation to the findings.

Response 9: Thank you for your professional vision. We are aware of this problem and have tried to find references. However, no high-quality references were found. If you have high-quality article recommendations, we will consider including them in the references of this article.

Comments 10: Conclusion. These are a bit vague, so please make it more concise and more relevant to the findings of the present study.

Response 10: Agree. Following your advice, we have rewritten the conclusion in lines 372 to 383, page 12.

4. Response to Comments on the Quality of English Language

Point 1: Numerous complicated sentences and grammatical errors are found in this manuscript.

Response 1: We appreciate your attention to the linguistic aspects of our manuscript. We understand that clear and concise communication is essential for scientific writing. we have carefully revised the language, paying attention to sentence structure, vocabulary choices, and tone. In addition, we also invited a friend of ours who is a native English speaker from the USA to help polish our article. We believe that these improvements will make our paper more accessible and engaging for readers.

Round 2

Reviewer 1 Report

Comments and Suggestions for Authors

Previous comment 8: The optical density of a material is defined as a logarithmic intensity ratio of the light falling upon the material, to the light transmitted through the material. Intensity of bands in Western blotting is “not” optical density.

Previous comment 2: Letters in Figures are still small and invisible after enlargement. Especially, captions to vertical axis of bar graphs in Figure 3, panel A, D, and Figure 5. Increase the resolution and/or font size. Letters in Figure 6 are also hard to recognize. Figure 4 is visible, readable, and appropriate size.

New comment 1: Check carefully references throughout the manuscript. For example; References 4 did not describe about fibronectin and integrin.  Reference 5 did not describe about Listeria.

 New comment 2: Abbreviation list: Rearrange alphabetical or appearance order.

 New comment 3: Use the same abbreviations as there are many different abbreviations for hematoxylin-eosin staining, such as “H&E staining”, “H. & E.”, “H&E.”, and “HE”. “HE” is seemed to be prefer.

Comments on the Quality of English Language

English must be polished up.

Author Response

Previous comment 8: The optical density of a material is defined as a logarithmic intensity ratio of the light falling upon the material, to the light transmitted through the material. Intensity of bands in Western blotting is “not” optical density.

Response 1: Agree. We totally agree with the problem you pointed out. We have made serious revisions to this issue on pages 6, 8, and 10.

Previous comment 2: Letters in Figures are still small and invisible after enlargement. Especially, captions to vertical axis of bar graphs in Figure 3, panel A, D, and Figure 5. Increase the resolution and/or font size. Letters in Figure 6 are also hard to recognize. Figure 4 is visible, readable, and appropriate size.

Response 2: Agree. Thank you very much for your suggestions. These problems occurred due to our negligence. We have modified the pictures on pages 6, 7, 8, 10 and 12 according to your suggestions.

New comment 1: Check carefully references throughout the manuscript. For example; References 4 did not describe about fibronectin and integrin.  Reference 5 did not describe about Listeria.

Response 3: Agree. Thank you very much for your careful review and we apologize for our oversight. We have checked the references section on pages 14-15 and revised the references with problems.

New comment 2: Abbreviation list: Rearrange alphabetical or appearance order.

Response 4: Agree. Thank you for pointing out this issue in the manuscript. We have modified the abbreviation list on page 13 according to your suggestion.

New comment 3: Use the same abbreviations as there are many different abbreviations for hematoxylin-eosin staining, such as “H&E staining”, “H. & E.”, “H&E.”, and “HE”. “HE” is seemed to be prefer.

Response 5: Agree. We have unified the abbreviations that appear multiple times in the manuscript, and according to your suggestion, we have changed them all to "HE".

4. Response to Comments on the Quality of English Language

Point 1: Numerous complicated sentences and grammatical errors are found in this manuscript.

Response 1: Thank you very much for your careful and responsible review of the manuscript. We understand that clear and concise communication is essential for scientific writing. We carefully checked the grammar and sentence problems in the manuscript and carefully revised the language, sentence structure, vocabulary, and tone. These improvements will make our manuscript more acceptable and attractive to readers. We thank you again for your attention to the language of our manuscript.

Reviewer 4 Report

Comments and Suggestions for Authors

No further issues in the manuscript after the careful revision by the authors.

Author Response

Thank you very much for your professional review of our article entitled “Activation of ARP2/3 and HSP70 Expression by Lipoteichoic Acid: Potential Bidirectional Regulation of Apoptosis in a Mastitis Inflammation Model” (ID: biomolecules-3061321). Based on the valuable opinions of the reviewers, we have revised the article. This will undoubtedly significantly improve the quality of our articles and make them more in line with the requirements of this journal.